# Loss of Nckx3 Exacerbates Experimental DSS-Induced Colitis in Mice through p53/NF-κB Pathway

**DOI:** 10.3390/ijms22052645

**Published:** 2021-03-05

**Authors:** Dinh Nam Tran, Seon Myeong Go, Seon-Mi Park, Eui-Man Jung, Eui-Bae Jeung

**Affiliations:** 1Laboratory of Veterinary Biochemistry and Molecular Biology, Veterinary Medical Center and College of Veterinary Medicine, Chungbuk National University, Cheongju, Chungbuk 28644, Korea; mr.tran90tb@gmail.com (D.N.T.); ksun3115@naver.com (S.M.G.); qkrtjsal0321@naver.com (S.-M.P.); 2Laboratory of Molecular Developmental Biology, Department of Molecular Biology, College of Natural Sciences, Pusan National University, Busandaehang-ro, 63beon-gil 2, Geumjeong-gu, Busan 46241, Korea; jungem@pusan.ac.kr

**Keywords:** calcium, Nckx3, colitis, inflammatory bowel disease

## Abstract

Inflammatory bowel diseases (IBDs) comprises a range of chronic inflammatory conditions of the intestinal tract. The incidence and prevalence of IBDs are increasing worldwide, but the precise etiology of these diseases is not completely understood. Calcium signaling plays a regulatory role in cellular proliferation. Nckx3, a potassium-dependent Na^+^/Ca^2+^ exchanger, is not only expressed in the brain but also in the aortic, uterine, and intestinal tissues, which contain abundant smooth muscle cells. This study investigated the role of Nckx3 in intestinal inflammation. Microarray analyses revealed the upregulation of the innate immune response-associated genes in the duodenum of Nckx3 knockout (KO) mice. The Nckx3 KO mice also showed an increase in IBD- and tumorigenesis-related genes. Using dextran sodium sulfate (DSS)-induced experimental colitis mice models, the Nckx3 KO mice showed severe colitis. Furthermore, the pathways involving p53 and NF-κB signaling were significantly upregulated by the absence of Nckx3. Overall, Nckx3 plays a critical role in the innate immune and immune response and may be central to the pathogenesis of IBD.

## 1. Introduction

Inflammatory bowel diseases (IBDs), consisting of ulcerative colitis (UC) and Crohn’s disease (CD), represent a group of idiopathic, chronic, inflammatory intestinal conditions [1]. The incidence of IBD has increased worldwide [2], affecting approximately 3.1 million persons in the United States [3]. The etiology of IBD has been associated with complex interactions among genetic, immune, and environmental factors, such as food intake and smoking [4]. On the other hand, the causative factors in the disease pathology are not completely understood, but animal and human studies have shown a strong genetic susceptibility.

Calcium (Ca^2+^) plays important roles in the physiological and biochemical functions of organisms and cells. Moreover, Ca^2+^ may be involved in modulating a wide array of cellular processes. Ca^2+^ signals can control the proliferation, differentiation, and function as well as a variety of transcriptional programs in immunocytes, such as T cells, B cells, mast cells, and many other cell types [5,6,7]. Furthermore, Ca^2+^ acts as a second messenger to regulate the innate immune cell function and activation [8]. At low Ca^2+^ concentrations, lymphocytes exist in a resting state. On the other hand, the engagement of antigen receptors induces calcium influx from the extracellular space and increases the intracellular Ca^2+^ concentrations [8]. In lymphocytes, intracellular Ca^2+^ concentrations regulate the activation of downstream signaling proteins, such as the calmodulin-dependent phosphatase calcineurin and calmodulin-dependent kinases, as well as downstream transcription factors, including NFAT (nuclear factor of activated T cells) [9]. Similarly, increased intracellular calcium levels play important roles in mediating the TLR-triggered immune response [10,11,12]. Thus, the calcium influx pathway has important roles in maintaining the intracellular Ca^2+^ concentration. IBD, particularly Crohn’s disease, which may impair calcium absorption [13]. Abnormalities in the intracellular Ca^2+^ stores contribute to the development of colonic dysmotility in UC and intestinal inflammation [14]. Furthermore, the calcium signaling pathway and hub genes, which are associated with immune dysregulation and tumorigenesis in various biological functions, may play an important role in UC development via multiple physiological and pathophysiological processes [15]. In contrast, supplementary calcium effectively prevents the increase in intestinal permeability and diminishes diarrhea in IBD [16]. Moreover, patients with IBD have a high risk of a reduced bone mineral density, which leads to osteopenia and osteoporosis [17,18]. On the other hand, calcium and vitamin D may prevent bone loss in osteoporotic patients with IBD [19]. Therefore, an examination of the calcium-signaling pathway may provide insight into the immunological and inflammatory response, which can help better understand IBD-based pathogenesis.

Plasma membrane sodium/calcium (Na^+^/Ca^2+^) exchangers play important roles in intracellular calcium homeostasis in various cells and organ systems. Two types of Na^+^/Ca^2+^ exchangers have been described in mammalian tissues, including the Na^+^/Ca^2+^ exchanger (NCX) family and the potassium-dependent sodium/calcium exchanger (Na^+^/Ca^2+^ K^+^; NCKX) family. In immune cells, both members of the NCX and NCKX families play a role in mast cell activation by controlling the sustained phase of Ca^2+^ mobilization [20]. Protein sodium/potassium/calcium exchanger 3 (Nckx3) is a member of the NCKX family that uses both the inward Na^+^ and outward K^+^ gradient to extrude Ca^2+^ [21]. Nckx3 was reported to be prominent in the hippocampus, cortex, and cerebellum in an adult brain. Moreover, transcripts of Nckx3 have been found in abundance in various other excitable body tissues with abundant smooth muscle, such as the aorta, intestine, lung, and uterus [22,23]. In intestinal tissue, Nckx3 is expressed in the basolateral membrane and plays a key role in calcium absorption [24]. Moreover, mRNA and protein levels of Nckx3 are high in the duodenum [25]. Thus, the abnormal expression of Nckx3 induces a decrease in calcium absorption. Recently, Nckx3 knockout (KO) mice showed a loss of bone mineral contents [26]. These studies indicate that Nckx3 may play a role in regulating calcium homeostasis. Furthermore, Nckx3 has been found in basophilic leukemia mast cells and murine bone marrow-derived mast cells. In these cells, the Ca^2+^ influx via Na^+^/Ca^2+^ exchanger was largely dependent on the NCKX family, which is in line with the prominent expression of Nckx3 [20]. Moreover, the NCKX family has been found to link with human pathologies, and mouse models [27]. However, the role of Nckx3 on human pathologies, and mouse models are unknown. In addition, Nckx3 was found on the basolateral membrane of the enterocytes in the intestine [28]. Thus, clarification on the role of Nckx3 is needed. This study examined the adult function of Nckx3 in resolving inflammation in a mouse model of colitis.

## 2. Results

### 2.1. Nckx3 KO Mice Exhibit Distinct Transcriptional Profiles of Proliferation- and Inflammation-Associated Genes

To investigate the possible interactions between Nckx3 and IBD, microarray analysis was performed to determine the differentially expressed genes in the duodenum of Nckx3 KO and wild-type (WT) mice. Affymetrix GeneChip CEL analysis showed that 192 genes were differentially expressed in Nckx3 KO mice. Figure 1A,B showed hierarchical clustering and box plots of the differentially expressed mRNAs. Among these differentially expressed genes, those for cell proliferation, tumorigenesis, immune response, and immune system processes were significantly different. As shown in Figure 1C, the expression levels of the cell proliferation-regulatory genes, including *Muc6*, *Cela1*, *Tff1*, *Tff2*, *Wfdc18*, and *Gkn3*, were significantly higher in Nckx3 KO mice. On the other hand, the levels of *Dnase1* and *Erdr1* expression were downregulated in Nckx3 KO mice (Figure 1C). Furthermore, the expression levels of 17 genes involved in initiating the immune response were markedly lower in the Nckx3 KO mice. The expression levels of immune system process-related genes were also significantly lower in the Nckx3 KO mice. These results suggest that Nckx3 loss may impair the immune systems in the intestine of mice. Quantitative RT-PCR was performed to verify the microarray results, as shown in Figure 2. We chose the top twelve most upregulated and downregulated mRNAs for validation.

### 2.2. Nckx3 Loss Impairs the Morphology and Cell Proliferation of the Gut

To determine the role of Nckx3 on the morphology and cell proliferation of the gut, the duodenum and colon were collected from 6-week-old Nckx3 KO mice and subjected to pathological analysis. Representative hematoxylin and eosin (H&E)-stained sections revealed the changes in the morphology of the duodenum (Figure 3A). The Nckx3 KO mice showed a shorter villus length than the WT mice (Figure 3A).

In the colon, there was no change in the morphology between the WT and Nckx3 KO mice (Figure 3B). The immunohistochemical analysis was performed to show the distribution of Nckx3 in the colons of Nckx3 KO and WT mice. As expected, Nckx3 staining was positive in WT and negative in Nckx3 KO mouse colonic tissues. Moreover, Nckx3 is expressed in the basolateral membrane as found in previous study [24]. This study next examined whether cell proliferation was affected in the Nckx3-deficient colon. Quantitation showed that the number of Ki67- positive cells was lower in the Nckx3 KO colon than in the WT colon (Figure 3D,E). Furthermore, the mRNA levels of cell proliferation-regulatory genes, including *Muc6*, *Muc5*, *Tff1*, and *Tff2*, were significantly higher in the colons of the Nckx3 KO mice than the WT mice (Figure 3F). These data show that the Nckx3 KO mutation affects the gross morphology of the intestines.

### 2.3. Nckx3 Loss Promotes Acute DSS-Induced Colitis

The microarray results and the change in the morphology and proliferation of intestines suggest that Nckx3 may play a role in the development of IBD. To further define the role of Nckx3 in IBD, the effects of Nckx3 loss in acute colitis were assessed by challenging the mice with dextran sodium sulfate (DSS), which induces experimental colitis with the clinical features of IBD. The WT and Nckx3 KO mice were exposed to 3% DSS in their drinking water. The DSS treatment induced a substantial weight loss in both WT and Nckx3 KO mice (Figure 4A). On the other hand, the Nckx3 KO mice exhibited significantly more weight loss than the WT mice from day 5 of the DSS treatment (Figure 4A). Furthermore, the WT mice showed more severe disease, as measured by the disease activity index score (Figure 4B). A decrease in colon length is a marker of intestinal inflammation. In this study, however, Nckx3 KO mice exhibited similar lengths to the equally treated WT mice (Figure 4C). Furthermore, histological scoring for inflammatory infiltrates and epithelial damage was higher in the Nckx3 KO mice than in the WT mice (Figure 4D). Overall, these results show that Nckx3 plays an important role in the development of colitis.

### 2.4. Nckx3 Deficiency Increased the Production of Dextran Sodium Sulfate (DSS)-Induced Proinflammatory Mediators

Several cytokines are regulated by NF- κB signaling. In this study, the mRNA levels of interleukin-6 *(IL-6*), *TNF-α, IFN- β*, *IL-1a* and *IL-1b* were higher in the colon of the DSS-treated groups than the non-treated DSS groups (Figure 5A). No significant differences in the mRNA levels of these genes were observed between the WT and Nckx3 KO mice (Figure 5A). On the other hand, the mRNA levels of these genes were significantly higher in the Nckx3 KO + DSS group than in the WT + DSS group (Figure 5A). The serum level of TNF-α was higher in the DSS-treated groups than the non-treated DSS groups (Figure 5B). Moreover, the group treated with DSS showed a higher serum level of IL-1a than the group not treated with DSS (Figure 5B). The serum level of TNF-α in the Nckx3 KO + DSS group was higher than that in the WT+DSS group (Figure 5B). The serum level of IL-1a in the Nckx3 KO + DSS group was also higher than that in the WT and Nckx3 KO groups (Figure 5B). The Nckx3 KO + DSS group exhibited a higher serum level of IL-1a than the WT + DSS group, but the difference was not significant (Figure 5B).

### 2.5. Nckx3 Loss Upregulated NF-κB Signaling in Acute DSS-Induced Colitis

The NF-κB signaling markers (p65) in the colon were analyzed by Western blotting to determine if Nckx3 can regulate the NF-κB activity (Figure 6). The protein levels of p65 in the WT and Nckx3 KO groups were similar (Figure 6A,B). On the other hand, treatment with DSS showed an increased level of p65 protein in the Nckx3 KO + DSS group compared to the WT, WT + DSS, and Nckx3 KO groups (Figure 6A,B). Moreover, the levels of p53 protein, a key tumor-suppressor gene, in the WT + DSS and Nckx3 KO + DSS groups were markedly higher than in the WT groups (Figure 6A,C). Furthermore, the Nckx3 KO mice also showed higher protein levels of p53 than the WT group (Figure 6A,C). There was no significant difference in the protein level of p53 between the WT + DSS and Nckx3 KO + DSS groups (Figure 6A,C).

The quantitative RT-PCR was performed to verify the expression of p65 and p53 in the colon of WT, WT + DSS, Nckx3 KO and Nckx3 KO + DSS. We found that the mRNA levels of p65 in the WT and Nckx3 KO groups were not significantly different (Figure 6D). However, WT + DSS and Nckx3 KO + DSS mice exhibited increased levels of p65 mRNA compared to the WT group (Figure 6D). Moreover, the level of p65 mRNA in the Nckx3 KO + DSS was markedly higher than in the WT + DSS, and Nckx3 KO groups (Figure 6D). Similarly, the mRNA level of p53 in the WT + DSS group was slightly higher than in the WT, WT + DSS, and Nckx3 KO groups (Figure 6D). The Nckx3 KO mice displayed higher mRNA levels of p53 than the WT group, but the difference was not significant (Figure 6D). In addition, treatment with DSS showed an increased level of p53 mRNA in the WT + DSS group compared to the WT group (Figure 6D). Overall, p53 is indispensable for Nckx3 KO-induced NF-κB signaling activation.

## 3. Discussion

Nckx3 plays a critical role in the development of IBD. These findings revealed an inverse correlation between Nckx3 and colitis. A deficiency of Nckx3 leads to severe mice colitis induced by DSS. Moreover, Nckx3 attenuates the inflammatory response by activating the expression of p53 and the NF-κB pathway.

DNA microarrays showed that the gene expression profiles of Nckx3 mice were slightly different from those of the WT mice. The levels of *Muc5ac* and *Muc6* mRNA expression, which are biomarkers of malignancy and chronic inflammation in the colonic mucosa, were significantly higher in the Nckx3 KO mice than the WT mice. MUC6 was not detected in normal or healthy ileal mucosae. On the other hand, MUC6 was expressed in many UC cases and correlated with the clinical markers associated with neoplasia [29]. MUC6 could also play an important role in the initial step of UC-associated tumorigenesis [30]. Moreover, MUC6 may play a role in epithelial wound healing after mucosal injury in inflammatory bowel diseases in addition to mucosal protection. It may also contribute to trefoil factors (TFFs) to epithelium restitution [31]. The coordinated localization of TFF2 and MUC6 was found in the stomach and duodenum. Furthermore, TFF2 and MUC6 were co-expressed in the same cells, shared the same secretory pathway and ended up in the same secretory vesicles [32]. TFF1, which belongs to the trefoil family, was expressed in severe UC patients but was not expressed in normal tissue [33]. TFF1 expression was also upregulated significantly in the duodenum from IBD patients [34]. Moreover, MUC5AC and TFF1 expression in goblet cells are common in IBDs and other inflammatory conditions of the colon [35]. The downregulation of Erdr1 leads to uncontrolled proliferation with low apoptotic properties [36]. In the intestine, Erdr1 was found in the crypt region and regulated the proliferation of crypt cells [37]. Recently, Erdr1 was reported to play an important role in the inflammatory process by regulating the immune system [38,39]. DNaseI, an endonuclease that facilitates chromatin breakdown and promotes susceptibility to autoimmune disorders, was lower in IBD patients [40]. In addition, the gel-forming mucins, MUC5AC and MUC6, as well as a variety of ions, including Ca^2+^, were lower in IBD patients [32]. The activity of DNase I required Mg^2+^ and Ca^2+^ cations [41].

Intestinal inflammation involves a complex interplay of innate and adaptive immune mechanisms. TNF-α is a major regulator of gel-forming mucins and plays a crucial role in regulating the innate immune responses. TNF-α has also been implicated in intestinal epithelial cell apoptosis in inflammatory diseases. Early studies showed that TNF-α is a marker of intestinal inflammation [42]. Furthermore, TNF is associated with the pathogenesis of IBD [43]. In IBD, the production of TNF was associated with growth failure in the relapse of the colon [44]. Moreover, TNF-α upregulated the expression of mucin secretion and increased the expression of MUC6, contributing to the defective mucus layer in colitis [45].

IFNs are crucial regulators of cell proliferation, differentiation, survival, and death, including type I (IFN-α, -β, -,ω or -τ), type II (IFN-γ), and type III (IFN-λ) [46]. Among these, type I IFNs are associated with multiple autoimmune and inflammatory disorders, including IBD [47]. Early studies revealed the important protective role of type I IFN in intestinal homeostasis under experimental colitis model conditions [48]. Moreover, type 1 IFN triggers an IFN-α/β response that may be of therapeutic value under intestinal inflammatory conditions [49]. In the present study, Nckx3 KO mice exhibited a decrease in the expression of the genes involved in interferon signaling (*IRF7*, *Oasl1*, *Oas1a*, *Itln1*, *Isg15*, *Ifit1*, *Ifi44*…). These results suggest that an Nckx3 deficiency induces the dysregulation of INF production and the response. Gene expression profiling analysis showed that a transcriptional network of inflammatory responses was downregulated significantly in the absence of Nckx3.

The disruption of the gut mucosal immune system can lead to a failure to induce protective immunity and maintain tolerance, which may result in severe gastrointestinal infections, IBD, or food allergies. Treatment with DSS mimics the condition of impaired immune tolerance because it disrupts the mucosal barrier of epithelial cells on the luminal surface. Studies of the DSS-induced colitis model system revealed the role of Nckx3 in driving the inflammatory response, which is not apparent in the non-stimulated state. DSS-treated Nckx3-deficient mice showed severe DSS colitis, which can be determined by several clinicopathological indicators, including more severe weight loss, diarrhea, hematochezia, shortening of the colon length, histological lesions, and increased proinflammatory cytokines. NF-κB, a central regulator of the innate immune response, plays a key role in the development of IBD [50]. The NF-κB-modulated infiltration of immune cells in the colon is believed to contribute to UC [51]. Furthermore, the expression and phosphorylation of NF-κB p65 was higher in a DSS-induced colitis model [52]. Moreover, NF-κB activating stimuli might equally require p53 for full NF-κB activation [53]. The tumor suppressor, p53, also has roles in inflammation and immunity [54]. Elevated p53 expression correlates with tumor progression [55]. Furthermore, p53 expression was closely associated with the development of UC with colorectal cancer [56]. Most studies described the opposing interactions between p53 and NF-κB, but these studies were conducted in either human cancer cells or mouse primary cells. In immune cells, p53 and NF-κB together drive the expression of cytokines and chemokines [57]. Indeed, the activation of p53 induced IL-6, TNF-α, and IL-8 expression [57]. Moreover, NF-κB was activated by various cytokines to activate the same and other proinflammatory cytokines and chemokines. The present study found that the levels of p53 protein were higher in the Nckx3 KO mice. This indicates the co-activation of p53 and p65 as important regulators of colitis in the absence of Nckx3. Furthermore, there is a synergistic relationship between the Ca^2+^ release pattern and the status of the p53 protein. An increase in Ca^2+^ was synchronized further with p53 transactivation, suggesting that Ca^2+^ signaling was responsible for p53 transactivation [58]. On the other hand, patients with p53 deletions exhibited higher serum calcium levels [59]. In the previous study, Nckx3 helped regulate calcium hemostasis [26]. Hence, Nckx3 may modulate the activation of p53 by regulating the calcium hemostasis.

The Nckx3 KO mice exhibited a change in the expression of the tumorigenesis-related genes. In clinical specimens, CASC4 was overexpressed in tumors and gastric carcinogenesis with Helicobacter pylori tissues [60]. Clusterin (CLU) plays a key role in the cell transformation process and has been reported to be overexpressed in several human tumor tissues, such as prostate, breast, renal, ovarian, and colon cancer [61,62]. Furthermore, the F5 gene is significantly upregulated in gastric cancer tumor tissues and may be a potential prognostic biomarker for gastric cancer [63]. ANXA10 represents a specific marker for adenocarcinomas of the upper gastrointestinal tract and pancreaticobiliary origin [64]. Upregulated *ANXA10* expression was also identified.

In summary, an Nckx3 deficiency increased p53 expression, cooperated with the activation of NF-κB inflammasome, thereby exacerbating intestinal inflammation in an experimental colitis model. The present study elucidated the pathogenesis of the intestinal inflammation associated with Nckx3, which might provide a novel therapeutic target. Future work will focus on the role of Nckx3 on the immune response.

## 4. Materials and Methods

### 4.1. Animals

For the generation of Nckx3 knock-out (KO) mice, zinc-finger nucleases technology was used as described in our previous study [26]. The mice were housed in polycarbonate cages with a 12 h light/dark cycle (lights on from 7:00 AM to 7:00 PM), a constant temperature of 23 °C, and humidity of 50%. Five mice were housed per cage and allowed access to water and food ad libitum. Mice were handled according to a protocol approved by the Institutional Animal Care and Use Committee of Chungbuk National University Institutional Animal Care and Use Committee (IACUC) (project identification code: CBNUR-1270-19-02).

### 4.2. Microarray Assay

For microarray, duodenums were isolated from WT and Nckx3 KO mice at 6 weeks of age. Then, RNAs were isolated using Trizol reagent (Invitrogen, Carlsbad, CA, USA) and prepared for Affymetrix mRNA microarray assay by Humanizing Genomics macrogen.

### 4.3. DSS-Induced Colitis

For DSS-induced colitis, the indicated mice (at 8 weeks) were administered 3% DSS in drinking water for 5 days, followed by 2 days of normal water. The body weight, stool consistency, and the presence of blood in the stool were recorded daily. Parameters for disease activity index (DAI) include weight loss (0, 0%; 1, 0–3%; 2, 3–6%; 3, 6–10%; 4, >10%), stool consistency (0, normal; 1, mildly; 2, very soft; 3, watery), and the presence of blood in the stool (0, normal; 1, brown; 2, reddish; 3, bloody stool). The combined DAI score ranged from 0 to 10.

### 4.4. Colitis Scores and Histologic Analysis

Colons were cut open longitudinally and fixed in 4% PFA followed by paraffin sectioning (5 µm thickness) and H&E staining. Briefly, 2 parameters were measured: inflammatory cell infiltrated (1, mucosa; 2, mucosa and submucosa; 3, transmural) and intestinal architecture (1, focal erosions; 2, erosions ± focal ulcerations; 3, erosions extended ulcerations ± granulation tissue ± pseudo polyps). The combined histopathologic score ranged from 0 to 6.

### 4.5. Immunofluorescence

Sections were deparaffinized, rehydrated and used for immunofluorescence stained. Sections were blocked with 5% goat serum (Vector Laboratories, Burlingame, CA, USA) for 1 h, followed by incubation in primary antibodies (Ki67, Cell Signaling Technology, Danvers, MA, USA, cat.no. D385, 1:500) at 4 °C overnight. For secondary staining, cells were incubated for 1 h in secondary antibody solution (Alexa Fluor488 goat anti-rabbit IgG, cat. no. A11034, 1:1000, Invitrogen, Carlsbad, CA, USA) that contain 100 ng/mL 4′,6- diamidino-2-phenylindole (DAPI) (Sigma-Aldrich). Then sections were mounted in Flouro-Gel (Emsdiasum, Hatfield, PA, USA).

### 4.6. RNA Extraction and Quantitative Real-Time PCR

Total RNA was extracted from duodenum and the colon of each mouse by using Trizol reagents (Ambion, Austin, TX, USA) according to the manufacturer’s protocols. cDNA synthesis was performed as previously described [65]. Quantitative real-time PCR analysis was carried out in a QuantStudio 3 (Applied Biosystems, Foster City, CA, USA). GAPDH served as internal control. The primer sequences are presented in Appendix A.

### 4.7. Western Blot Analysis

Protein was extracted from colon from the WT and Nckx3 KO mice at 2 days after DSS treatment using pro-prep solution (iNtRON, Seoul, Korea) according to the manufacturer’s protocol. Then, 100 µg of protein was separated on 12% sodium dodecyl sulfate–polyacrylamide gel electrophoresis and transferred to polyvinylidene fluoride membrane (Merck Millipore, Taunton, MA, USA) as previously described [66]. After blocking with 5% skim milk in TBS with 0.05% Tween-20, the membrane was incubated overnight in primary antibodies 4 °C (p65, Santa Cruz Biotechnology, Santa Cruz, CA, USA, cat. no. sc-7151, 1:500; p53, Santa Cruz Biotechnology, cat. no. sc-126, 1:500) and secondary antibodies (anti-rabbit, Cell Signaling Technology, cat. no. 7074S, 1:3000; anti-mouse, Cell Signaling Technology, cat. no. 7076P2, 1:3000). Membranes were enhanced using chemiluminescence reagent (EMD Millipore Corporation, Burlington, MA, USA). The band intensities of the target proteins were detected with the Chemi Doc equipment, GenGnome5 (Syngene, Cambridge, UK) and analyzed by using Image J software and normalized by the band intensity of GAPDH (Cell Signaling Technology, MA, USA).

### 4.8. ELISA Assay

Blood samples were isolated from WT and Nckx3 KO mice at 6 weeks of age. Then, the serum was collected, and cytokines were detected using ELISA kit (TNF-a ELISA kit and IL-4 ELISA kit, Cloud-clone Corp., MD, USA) according to the manufacturer’s instructions.

### 4.9. Statistical Analysis

Data were analyzed using GraphPad Prism software (GraphPad Software, La Jolla, CA, USA) and presented as the means ± SEM. The *p* values for each comparison are described in the results section. Statistical significance was determined by two two-way ANOVA, unpaired Student’s t tests for two population comparisons or one-way ANOVA (Bonferroni’s multiple comparison test). The allocation, treatment, and handling of animals were the same across study groups.

## Figures and Tables

**Figure 1 ijms-22-02645-f001:**
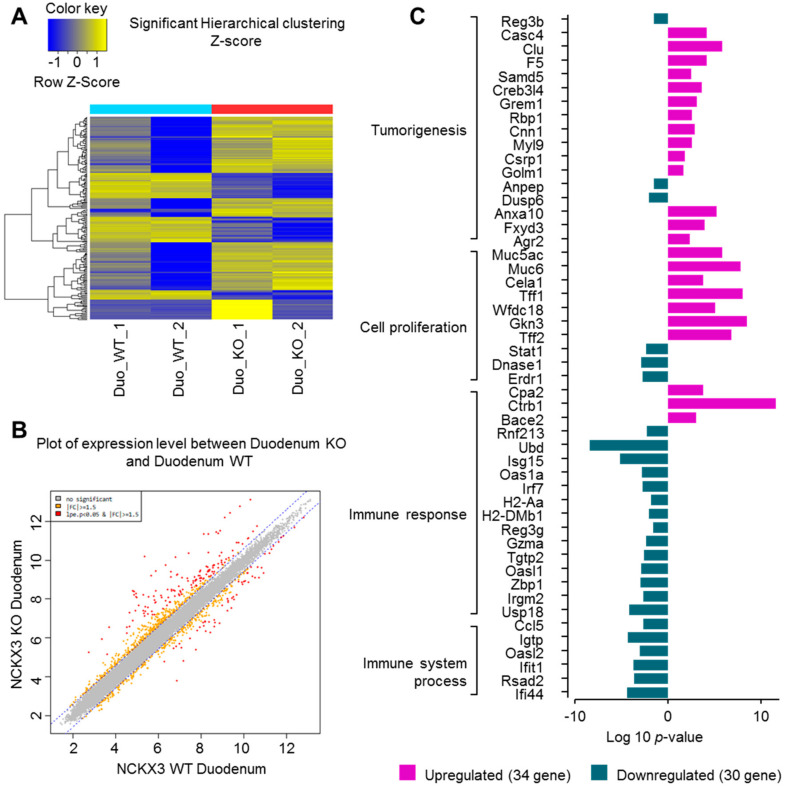
Gene expression profile in duodenum upon the genetic deletion of Nckx3 locus. mRNA was collected from duodenum tissue of wild-type (WT) and Nckx3 knockout (KO) mice at 6-week-aged, followed by Affymetrix mRNA microarray assay: (**A**) differentially expressed mRNAs were analyzed using hierarchical clustering. ‘Blue’ indicates low relative expression and ‘yellow’ indicates the high relative expression, and ‘blue’ indicates a low relative expression; (**B**) box plot showing the distribution of maximum, minimum, and percentile values for the normalized signal of each sample; (**C**) curated list of significantly enriched gene ontology (GO) terms in the lists of up and downregulated genes.

**Figure 2 ijms-22-02645-f002:**
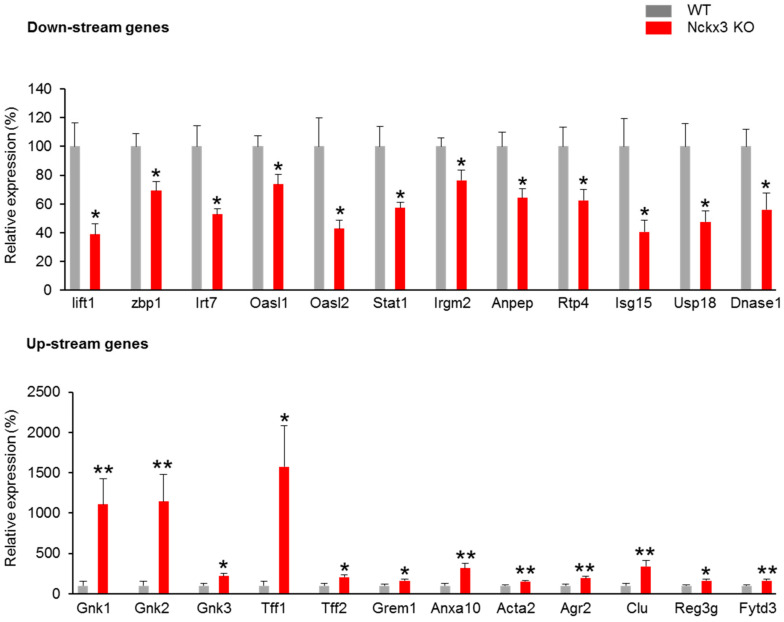
mRNA levels of the top twelve most up and downstream genes were assessed in WT and Nckx3 KO duodenum tissues using real-time PCR (n = 10 per genotype). Data represent the mean ± SEM. Statistical significance was determined by two-way ANOVA, * *p* < 0.05 WT vs. Nckx3 KO, ** *p* < 0.01 WT vs. Nckx3 KO.

**Figure 3 ijms-22-02645-f003:**
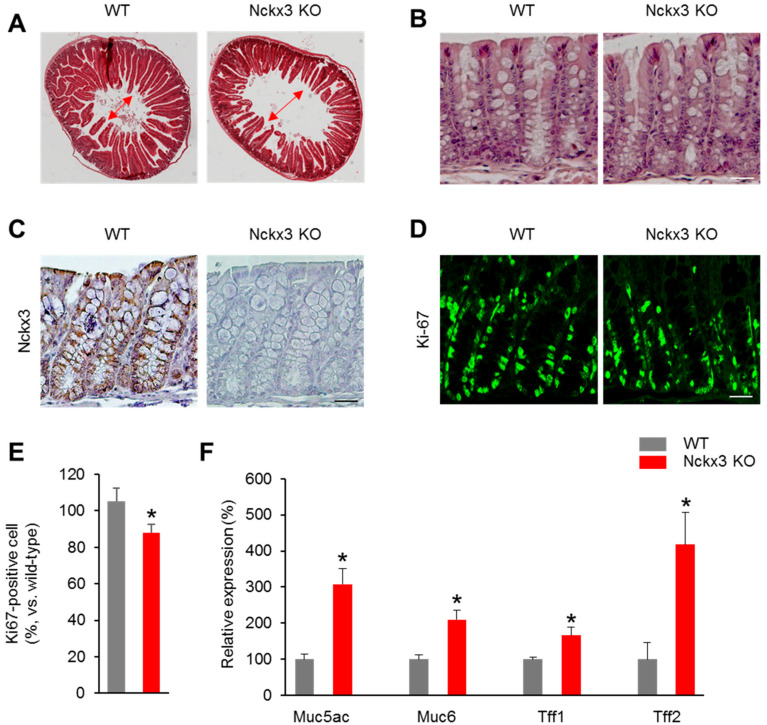
Loss of Nckx3 impairs the morphology in the duodenum and cell proliferation in the colon. Hematoxylin and eosin (H&E)-stained sections of duodenum and middle distal colon tissue collected from WT and Nckx3 KO mice at 6-week-aged (n = 10 per genotype): (**A**) a markable decrease in the length of crypt in the duodenum of Nckx3 KO mice compared to the WT mice. Original magnification: 4×. However, there was no significant in the morphology of crypt morphology between WT and Nckx3 KO mice (**B**). Original magnification: 10×; (**C**) immunohistochemistry analysis of Nckx3 expression in colonic tissues of WT and Nckx3 KO mice—original magnification: 10×; (**D**) immunofluorescence detection of Ki67 on colon sections prepared from WT and Nckx3 KO mice—original magnification: 10×; (**E**) quantification of (**D**). The number of Ki67^+^ cells was decreased in the Nckx3 KO group compared to WT group; (**F**) Nckx3 KO groups showed an increase in the mRNA level of proliferation-regulatory genes including *Muc5a*, *Muc6*, *Tff1* and *Tff2* in the colon compared to the WT group (n = 10 per genotype). Data represent the mean ± SEM. Statistical significance was determined by two-way ANOVA, * *p* < 0.05 WT vs. Nckx3 KO.

**Figure 4 ijms-22-02645-f004:**
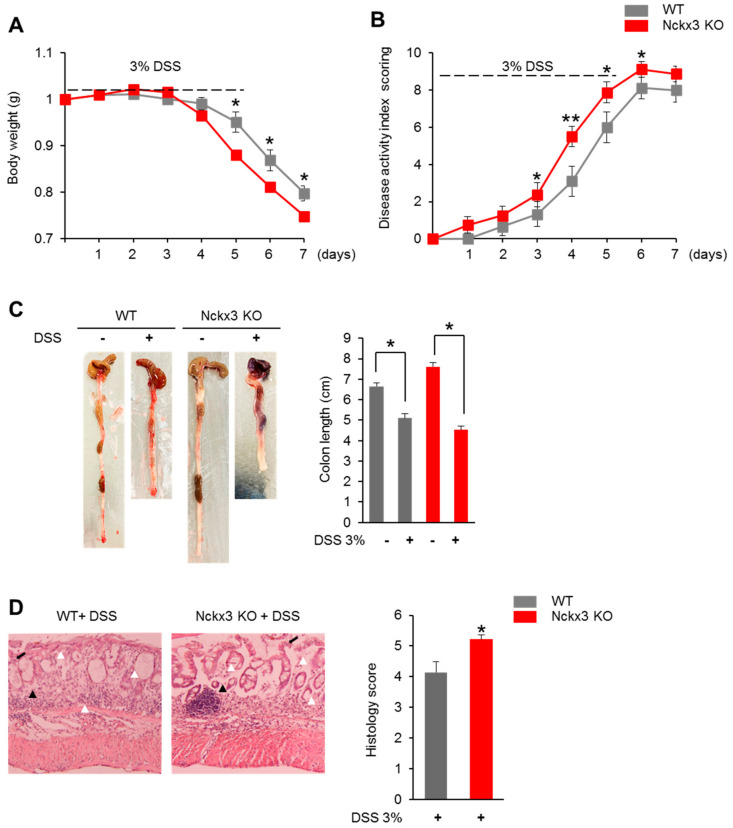
Nckx3 deficiency enhanced the severity of dextran sulfate sodium (DSS)-induced colitis. Colitis was induced in WT and Nckx3 KO mice by the administration of 3% DSS in the drinking water for 5 days followed by regular drinking water for an additional 2 days. Mice were evaluated daily for (**A**) weight loss and (**B**) disease activity index (DAI) scores were calculated. Colon length was also measured (**C**). Representative colon histological sections stained with hematoxylin and eosin (H&E) were shown at (**D**) days 7 after colitis induction (n = 10 per group). Images were shown at original magnification 10X. Histological sections were blindly scored on a scale of 0 to 6 to generate a histological score and individual mouse scores were shown with each data point representing a single mouse. Arrow indicates the ulceration; white arrowhead: neurotrophil; black arrowhead: polyps. The combined histopathologic score ranged from 0 to 6. Nckx3 KO + DSS group exhibited higher in the histological scoring for inflammatory infiltrates and epithelial damage than in the WT group. Data represent the mean ± SEM. Statistical significance was determined by two-way ANOVA, * *p* < 0.05 WT +DSS vs. Nckx3 KO + DSS.

**Figure 5 ijms-22-02645-f005:**
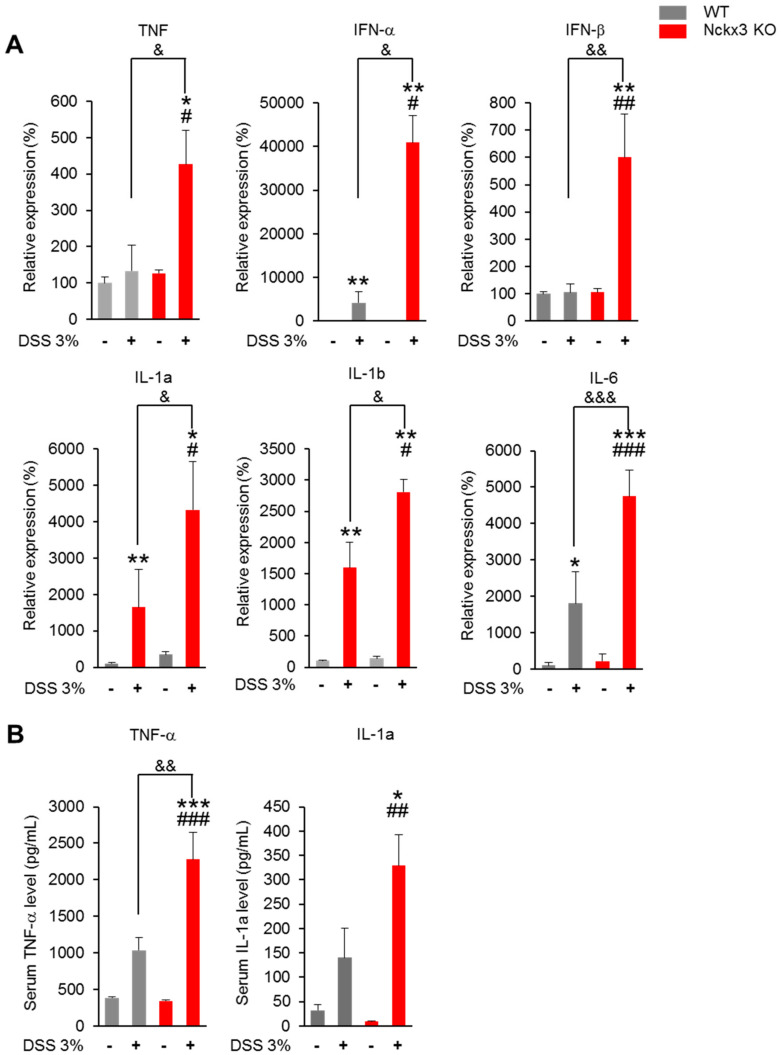
Nckx3 upregulated NF-κB signaling in colitis mice. (**A**) Compared with the WT + DSS group, the Nckx3 KO + DSS group exhibited increased *IL-6*, *TNF-α*, *IL-1β* and *IL-17a* mRNA expression levels in the colon; (**B**) compared with the WT + DSS group, the Nckx3 KO + DSS group exhibited increased TNF-α and IL-1a concentrations in serum (n = 10 per genotype). Data represent the mean ± SEM. Statistical significance was determined by one-way ANOVA with Bonferroni correction. * *p* < 0.05 vs. WT, ** *p* < 0.01 vs. WT Nckx3 KO, ** *p* < 0.001 vs. WT, *** *p* < 0.001 vs. WT, ^#^
*p* < 0.05 Nckx3 KO + DSS vs. Nckx3 KO, ^##^
*p* < 0.01 Nckx3 KO + DSS vs. Nckx3 KO, ^###^
*p* < 0.001 Nckx3 KO + DSS vs. Nckx3 KO, ^&^
*p* < 0.05 WT + DSS vs. Nckx3 KO + DSS, ^&&^
*p* < 0.01 WT + DSS vs. Nckx3 KO + DSS, ^&&&^
*p* < 0.001 WT + DSS vs. Nckx3 KO + DSS.

**Figure 6 ijms-22-02645-f006:**
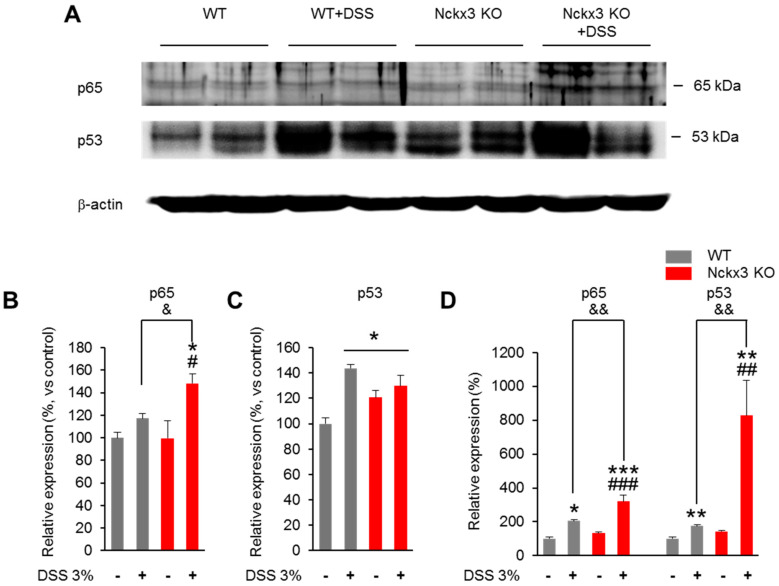
Nckx3 regulates NF-κB activation by targeting p53. (**A**) NF-κB p65 and p53 and GAPDH protein expression levels in the colon of WT and Nckx3 KO mice were detected by Western blotting. (**B**,**C**) Quantification of (**A**). The protein level of p65 was increased in the WT + DSS compared to the WT group, but not significantly. However, the Nckx3 KO +DSS groups showed a markedly higher protein level of p65 than in the WT, WT + DSS and Nckx3 KO groups. In addition, the WT, WT + DSS and Nckx3 KO groups showed higher protein levels of p53 than the WT group (n = 10 per group). (**D**) RT-qPCR analysis NF-κB p65 and p53 mRNA expression levels in colon of WT and Nckx3 KO mice (n = 10 per genotype). Data represent the mean ± SEM. Statistical significance was determined by one-way ANOVA with Bonferroni correction. * *p* < 0.05 vs. WT, ** *p* < 0.01 vs. WT, *** *p* < 0.001 vs. WT, ^#^
*p* < 0.05 Nckx3 KO + DSS vs. Nckx3 KO, ^##^
*p* < 0.01 Nckx3 KO + DSS vs. Nckx3 KO, ^###^
*p* < 0.01 Nckx3 KO + DSS vs. Nckx3 KO, ^&^
*p* < 0.05 WT + DSS vs. Nckx3 KO + DSS, ^&&^
*p* < 0.01 WT + DSS vs. Nckx3 KO + DSS.

## Data Availability

The datasets generated during and/or analyzed during the current study are available from the corresponding author on reasonable request.

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
