# Peer review of "Loss of Nckx3 Exacerbates Experimental DSS-Induced Colitis in Mice through p53/NF-κB Pathway"

_ijms, 2021, doi:10.3390/ijms22052645_

Round 1

Reviewer 1 Report

In the manuscript entitled “Loss of Nckx3 exacerbates experimental DSS-induced colitis in mice through p53/NF-kB pathway”, the authors illustrated that Nckx3 plays a critical role in the innate immune and immune response and may be central to the pathogenesis of IBD. However, the manuscript in its current form should be considered for publication in a more specialized journal.

  1. What are the rationales to use Nckx3 as the target? The authors can use the public database to compare the patient and healthy people’Nckx3 level in IBD.
  2. For figure 1, please include Hierarchical clustering and Volcano analysis to support the results.
  3. In figure 2A, the markers cause confusion. Please simplify the style.
  4. Scale bars were missed in figure 3b and c.
  5. Why are Muc5ac and Muc6 not included in figure 2, while Tiff1 and 2 are?
  6. In figure 4a, the authors show 5 days 3% DSS and 2 days water. How are the mice body weights unchanged in the first 5 days with DSS but decreased later when just drinking water?
  7. Figure 4a,b,d need data from 4 animal groups in total.
  8. In figure 6, just using WB to demonstrate that Nckx3 regulated NF-kB activation by targeting p53 is not convincing. Please use more experiments to support these results.
  9. In the abstract and figure 1 Gene expression profile, immune response differences are mentioned. However, in the rest of the figures, only cell proliferation and inflammation data are shown. The authors should narrow the focus and and provide more supporting data.

Reviewer 2 Report

This manuscript reports that Nckx3 KO mice developed severe colitis induced by 3% DSS in drinking water, when compared to the wild mice. The authors also found that the pathways involving p53 and NF-kappaB signaling were significantly upregulated by the absence of Nckx3. They concluded that Nckx3 plays a critical role in the innate immune and immune response and may be central to the pathogenesis of IBD. The manuscript contains some interesting findings. However, this reviewer has several concerns.

1) Organ‑specific protein expression of Nckx3 should be described. In the Ref. 25, mRNA and protein levels of Nckx3 are high in the duodenum, but not in colon. Also, such distribution in the colon should be added, because DSS can affect the distal colon. In this study, the authors are recommended to add the immunohistochemical distribution of Nckx3 in the colons of Nckx3 KO and wild mice.

2) Are there reports on the involvement of Nckx3 in the pathogenesis/pathobiology of human IBD?

3) Some sentences (page 11, the 2nd paragraph) are unclear.

Round 2

Reviewer 1 Report

This work is interesting and the results are also beneficial for identifying Loss of Nckx3 exacerbates experimental DSS-induced colitis in mice through p53/NF-kB pathway. No further comment on the revised manuscript.

Author Response

We would like to thank you for your comment.

Reviewer 2 Report

The revised manuscript has been improved. Just one thing, Figure 3 is duplicated. Please delete one.
